# OBJECT PURSUIT: BUILDING A SPACE OF OBJECTS VIA DISCRIMINATIVE WEIGHT GENERATION

**Chuanyu Pan[1,*], Yanchao Yang[2,*], Kaichun Mo[2], Yueqi Duan[1,2], Leonidas Guibas[2]**
[1]Tsinghua University    [2]Stanford University
pancy17@mails.tsinghua.edu.cn
{yanchaoy, kaichun, guibas}@cs.stanford.edu
duanyueqi@tsinghua.edu.cn

## ABSTRACT

We propose a framework to continuously learn object-centric representations for visual learning and understanding. Existing object-centric representations either rely on supervisions that individualize objects in the scene, or perform unsupervised disentanglement that can hardly deal with complex scenes in the real world. To mitigate the annotation burden and relax the constraints on the statistical complexity of the data, our method leverages interactions to effectively sample diverse variations of an object and the corresponding training signals while learning the object-centric representations. Throughout learning, objects are streamed one by one in random order with unknown identities, and are associated with latent codes that can synthesize discriminative weights for each object through a convolutional hypernetwork. Moreover, re-identification of learned objects and forgetting prevention are employed to make the learning process efficient and robust. We perform an extensive study of the key features of the proposed framework and analyze the characteristics of the learned representations. Furthermore, we demonstrate the capability of the proposed framework in learning representations that can improve label efficiency in downstream tasks. Our code and trained models are made publicly available at: `https://github.com/pptrick/Object-Pursuit`.

## 1 INTRODUCTION

What are human infants and toddlers learning while they are manipulating a discovered object? And, how do such continual interaction and learning experiences, i.e., objects are discovered and learned one by one, help develop the capability to understand the scenes that consist of individual objects? Inspired by these questions, we aim for training frameworks that enable an autonomous agent to continuously learn object-centric representations through self-supervised discovery and manipulation of objects, so that the agent can later use the learned representations for visual scene understanding.

A majority of object-centric representation learning methods focus on encoding images or video clips into disentangled latent codes, each of which explains an entity in the scene, and together they should reconstruct the input. However, without explicit supervision and more sophisticated inductive biases beyond parsimony, the disentanglement usually has difficulties aligning with objects, especially for complex scenes. We leverage the fact that an autonomous agent can actively explore the scene, and propose that the data collected by manipulating a discovered object can serve as an important source for building inductive biases for object-level disentanglement.

In our proposed framework, whenever an object is discovered by the agent, a dataset containing images and instance masks of this object can easily be sampled via interaction compared to annotating all the objects. Theoretically speaking, any function of the images induced by the discovered object could be a representation of the object. For example, let $\phi$ be an encoder implemented by a neural network, and let x be the image of an object, we can say that $\phi(\mathrm{x})$ is a representation of the object. Similarly, the encoder itself can also be a representation of this object since $\phi = \arg\min_\phi \mathcal{L}(\phi, \mathrm{x})$, i.e., $\phi$ is the output of an optimization procedure that takes the object's images as input.

---

[*]Equal Contribution.

We employ network weights as the object-centric representations. Specifically, the proposed method learns an object-centric representation from the data collected by manipulating a single object, through learning a latent code that can be translated into a neural network. The neural network is produced by a discriminative weight generation hypernetwork and is able to distinguish the represented object from anything else. In order to learn representations for objects that stream in one by one, the proposed framework is augmented with an object re-identification procedure to avoid learning seen objects. Moreover, we hypothesize that object representations are embedded in a low-dimensional manifold, so the proposed framework first checks whether a new object can be represented by learned objects; if not, the new object will be learned as a base object serving the purpose of representing future objects, thus the name *object pursuit*. Furthermore, the proposed framework deals with the catastrophic forgetting of learned object representations by enforcing the hypernetwork to maintain the mapping between the learned representations and their corresponding network weights.

In summary, our work makes the following contributions: 1) we propose a novel framework named *object pursuit* that can continuously learn object-centric representations using training data collected from interactions with individual objects, 2) we perform an extensive study to understand the pursuit dynamics and characterize its typical behaviors regarding the key design features, and 3) we analyze the learned object space, in terms of its succinctness and effectiveness in representing objects, and empirically demonstrate its potential for label efficient visual learning.

## 2 RELATED WORK

**Object-centric representation learning** falls in the field of disentangled representation learning (Higgins et al., 2016; Kim & Mnih, 2018; Press et al., 2019; Chen et al., 2018b; Karras et al., 2019; Li et al., 2020; Locatello et al., 2020a; Zhou et al., 2021). However, object-centric representations require that the disentangled latents correspond to objects in the scene. For example, (Eslami et al., 2016; Kosiorek et al., 2018) model image formation as a structured generative process so that each component may represent an object in the generated image. One can also apply inverse graphics (Yao et al., 2018; Wu et al., 2017) or spatial mixture models (Greff et al., 2017; 2019; Engelcke et al., 2020b) to decompose images into interpretable latents. Monet (Burgess et al., 2019) jointly predicts segmentation and representation with a recurrent variational auto-encoder. Capsule autoencoders (Kosiorek et al., 2019) are proposed to decompose images into parts and poses that can be arranged into objects. To deal with complex images or scenes, (Yang et al., 2020; Bear et al., 2020) employ motion to encourage deomposition into objects. Besides motion, (Klindt et al., 2021) shows that the transition statistics can be informative about objects in natural videos. Similarly, (Kabra et al., 2021) infers object latents and frame latents from videos. Slot-attention (Locatello et al., 2020b; Jiang et al., 2020) employs the attention mechanism that aggregates features with similar appearance, while Giraffe (Niemeyer & Geiger, 2021) factorizes the scene using neural feature fields. Even though better performance is achieved with more sophisticated network designs, scenes with complex geometry and appearance still lag. As shown in (Engelcke et al., 2020a), the reconstruction bottleneck has critical effects on the disentanglement quality. Instead of relying on reconstruction as a learning signal, our work calls for interactions that stimulate and collect training data from complex environments.

**Rehearsal-based continual learning.** In general, continual learning methods can be divided into three streams: rehearsal-based, regularization-based, and expansion-based. The rehearsal-based method manages buffers to replay past samples, in order to prevent from forgetting knowledge of the preceding tasks. The regularization-based methods learn to regularize the changes in parameters of the models. The expansion-based methods aim to expand model architectures in a dynamic manner. Among these three types, rehearsal-based methods are widely-used due to their simplicity and effectiveness (Lüders et al., 2016; Kemker & Kanan, 2017; Rebuffi et al., 2017; Cha et al., 2021; von Oswald et al., 2019; Riemer et al., 2018; Lopez-Paz & Ranzato, 2017; Buzzega et al., 2020; Aljundi et al., 2019; Chaudhry et al., 2020; Parisi et al., 2018; Lopez-Paz & Ranzato, 2017). Samples from previous tasks can either be the data or corresponding network activations on the data. For example, (Shin et al., 2017) proposes a dual-model architecture where training data from learned tasks can be sampled from a generative model and (Draelos et al., 2017; Kamra et al., 2017) propose sampling in the output space of an encoder for training tasks relying on an auto-encoder architecture. ICaRL Rebuffi et al. (2017) allows adding new classes progressively based on the training samples

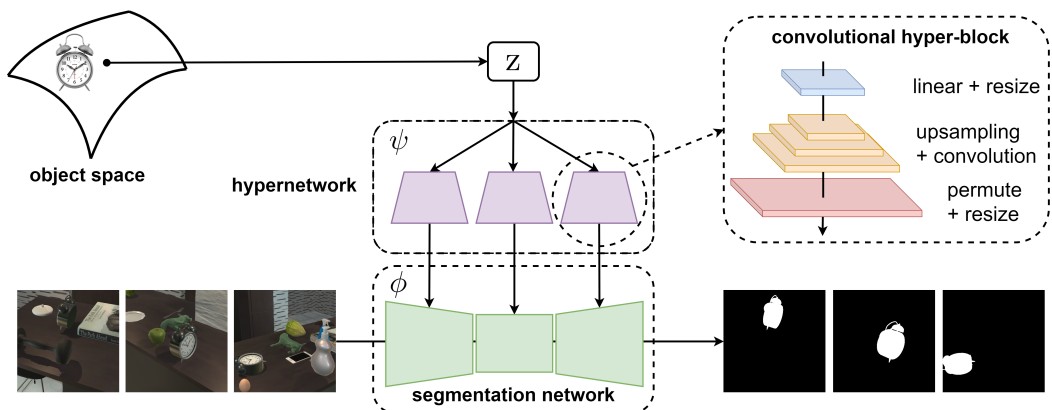

Figure 1: Object space as discriminative weights. Objects live in a low-dimensional manifold of a high-dimensional latent space. A latent code representing a specific object is translated into segmentation weights that can distinguish the object from anything else at different viewing conditions. The hypernetwork consists of blocks built of convolutional and upsampling layers.

with a small number of classes, while (Pellegrini et al., 2020; Li & Hoiem, 2017) store activations volumes at some intermediate layer to alleviate the computation and storage requirement. $Co^2L$ (Cha et al., 2021) proposes continual learning within the contrastive representation learning framework, and (Balaji et al., 2020) studies continual learning in large scale where tasks in the input sequence are not limited to classification. Similar to the forgetting prevention component in our framework, von Oswald et al. (2019) applies a task-conditioned hypernetwork to rehearse the task-specific weight realizations. Please refer to (Parisi et al., 2019; Delange et al., 2021) for a more comprehensive review on this subject.

**Hypernetwork.** The goal of hypernetworks is to generate the weights of a target network, which is responsible for the main task (Ha et al., 2016; Krueger et al., 2017; Chung et al., 2016; Bertinetto et al., 2016; Lorraine & Duvenaud, 2018; Sitzmann et al., 2020; Nirkin et al., 2021). For example, (Krueger et al., 2017) proposes Bayesian hypernetworks to learn the variational inference in neural networks and (Bertinetto et al., 2016) proposes to learn the network parameters in one shot. Hyper-Seg (Nirkin et al., 2021) presents real-time semantic segmentation by employing a U-Net within a U-Net architecture, and (Finn et al., 2019) applies hypernetwork to adapt to new tasks for continual lifelong learning. Moreover, (Tay et al., 2020) proposes a new transformer architecture that leverages task-conditioned hypernetworks for controlling its feed-forward layers, whereas (Ma et al., 2021) proposes hyper-convolution, which implicitly represents the convolution kernel as a function of kernel coordinates. Hypernetworks have shown great potential in different meta-learning settings (Rusu et al., 2018; Munkhdalai & Yu, 2017; Wang et al., 2019), mainly due to that hypernetworks are effective in compressing the primary networks' weights as proved in (Galanti & Wolf, 2020).

## 3 METHOD

We consider an agent that can explore the environment and manipulate objects which are discovered in an unknown order. Suppose there are $N$ objects in the scene, each of which randomly appears in an image $x \in \mathbb{R}^{H \times W \times 3}$, whose ground-truth instance segmentation mask is $y \in \mathbb{R}^{H \times W \times N}$. One can train a deep neural network that maps an image $x$ to its mask $y$ with a dataset $\mathcal{D} = \{(x_i, y_i)\}$ that consists of such paired training samples. However, sampling from the joint distribution $p(x, y)$ can be extremely time-consuming, e.g., someone may have to manually draw the instance masks for every object in an image.

On the other hand, sampling from the marginals can be much more accessible through interactions. Let $\mathcal{D}^k$ be the dataset collected by observing an image $x_i$ and the corresponding binary mask of the $k$-th object $y_i^k \in \mathbb{R}^{H \times W}$, i.e., $\mathcal{D}^k = \{(x_i, y_i^k)\} \sim p(x, y^k)$, which is the marginal distribution obtained by integrating out other objects' masks in $y$. The goal of the proposed object pursuit framework is to learn object-centric representations from the data collected by continuously sampling the marginals. Next, we detail the representations used for objects (as illustrated in Fig. 1), and how we can learn them without catastrophic forgetting.

## 3.1 Representing Objects via Discriminative Weight Generation

In order to represent an object, one can compute any functions of the data produced with this object. For example, the encoding of an image containing a specific object that can be used to reconstruct the input image. Here we take a conjugate perspective instead of asking the representation to store information of an object that is good for reconstruction. We propose that the object-centric representation of an object shall generate the mechanisms for performing certain downstream tasks on this object, e.g., distinguishing this object from the others.

Let $\phi$ be a segmentation network with learnable weights $\theta$ that maps an image to a binary mask, i.e., $\phi : \Theta \times \mathbb{R}^{H \times W \times 3} \to \mathbb{R}^{H \times W}$. Moreover, let $\psi : \zeta \to \Theta$ be the mapping from the latent space $\zeta$ to the weights of the segmentation backbone $\phi$. We define the object-centric representation of an object $o$ as a latent $z_o \in \zeta$, such that:

$$\mathbb{E}_{(x_i, y_i^o) \sim p(x, y^o)} \Delta(\phi(\psi(z_o), x_i), y_i^o) \geq \tau, \tag{1}$$

where the expectation is computed according to $p(x, y^o)$, i.e., the marginal distribution of object $o$, and $\Delta$ is a similarity measure between the prediction from $\phi$ and the sampled mask $y^o$. In other words, $z_o$ is a representation of object $o$, if the network weights generated from $z_o$ are capable of predicting high-quality instance masks regarding the object under the corresponding marginal distribution. The threshold $\tau$ is a scalar parameter that will be studied in the experiments. Now we detail the proposed *object pursuit* framework, which unifies object re-identification, succinctness of the representation space, and forgetting prevention, for continuously learning object representations.

## 3.2 Object Pursuit

Given the definition of object-centric representations in Eq. 1, our goal is to construct a low-dimensional manifold to embed objects in the input space $\zeta$ of the weight generation hypernetwork $\psi$. We conjecture that the low-dimensional manifold can be spanned by a set of base object representations. More explicitly, we instantiate two lists $\mathbf{z}$ and $\boldsymbol{\mu}$, which store the representations of the base objects and the embeddings of the learned objects, respectively. We denote $\mathbf{z}^{t-1} = \{z_i\}_{i=1}^m$ and $\boldsymbol{\mu}^{t-1} = \{\mu_i\}_{i=1}^n$ ($n \geq m$, with $n$ the number of learned objects and $m$ the number of base objects, up to time $t-1$) as the constructed lists after encountering a $(t-1)$-th object. Note that $\mu_i$ has the same dimension as the number of base object representations. Similarly, we denote $\psi^{t-1}$ as the corresponding hypernetwork parameters.

As discussed, when the $t$-th object $o_t$ is discovered, a dataset $\mathcal{D}^t = \{(x_j, y_j^t)\}$ can be easily sampled from the marginal distribution $p(x, y^t)$ through interactions. However, such object might already be seen previously. Thus, it is necessary to apply re-identification to avoid repetitively learning the same object. According to the definition in Eq. 1, object $o_t$ will be claimed as a seen or learned object if the following condition is true ($|\cdot|$ is the cardinality of a set):

$$\max_{i \leq |\boldsymbol{\mu}^{t-1}|} \mathbb{E}_{(x_j, y_j^t) \in \mathcal{D}^t} \Delta(\phi(\psi^{t-1}(z_i), x_j), y_j^t) \geq \tau. \tag{2}$$

with $z_i = \mu_i \cdot \mathbf{z}^{t-1}$. In this case, object $o_t$ will be assigned the identity $i^*$ that achieves the maximum value. Otherwise, if Eq. 2 is not valid, $o_t$ is considered as an object that has not been learned.

**Learning base object representations.** An object $o_t$ that can not be identified with the list of learned objects $\boldsymbol{\mu}^{t-1}$ can potentially serve as a base object whose representation should be added to the list of base representations $\mathbf{z}$. To ensure that object $o_t$ qualifies as a base object, we propose the following test which checks whether $o_t$ can be embedded in the current manifold spanned by $\mathbf{z}^{t-1}$:

$$\mu^* = \arg\max_{\mu \in \mathbb{R}^{|\mathbf{z}^{t-1}|}} \mathbb{E}_{(x_j, y_j^t) \in \mathcal{D}^t} \Delta(\phi(\psi^{t-1}(\mu^T \mathbf{z}^{t-1}), x_j), y_j^t) + \alpha \|\mu\|_1, \tag{3}$$

where $\mu^*$ is the optimal embedding for object $o_t$ regarding $\mathbf{z}^{t-1}$ under the $\ell_1$ regularizer to encourage sparsity. If the first term of Eq. 3 passes the threshold $\tau$ with the representation $\mu^{*T} \mathbf{z}^{t-1}$, then we consider $o_t$ as an object that should not be added to the list of bases since it can already be represented by the existing base objects.

Next, if $o_t$ does not fall on the manifold spanned by $\mathbf{z}^{t-1}$, a joint learning of the representation of $o_t$ and the hypernetwork $\psi$ shall be performed so that a new base object representation can be added

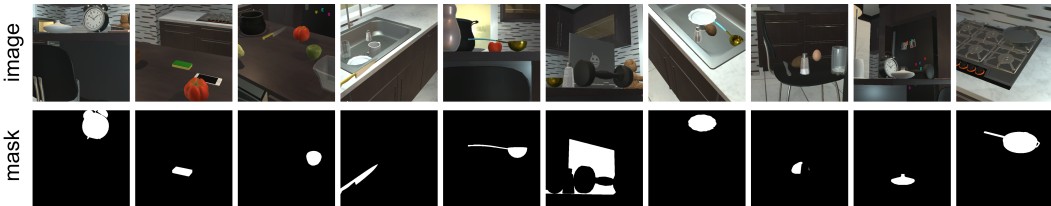

Figure 2: Data collected in iThor. Target objects are highlighted by their instance masks.

to the list. However, since updating the hypernetwork could result in catastrophic forgetting of the previously learned object representations, it is also necessary to constrain the learning process, and the training loss is:

$$z^*, \psi^* = \arg\max_{z, \psi} \mathbb{E}_{(x_j, y_j^t) \in \mathcal{D}^t} \Delta(\phi(\psi(z), x_j), y_j^t) + \alpha \|z\|_1$$
$$+ \beta \sum_{i \leq |\boldsymbol{\mu}^{t-1}|} \|\psi(\mu_i^T z^{t-1}) - \psi^{t-1}(\mu_i^T z^{t-1})\|_1, \quad (4)$$

where the first two terms help to find a good representation for object $o_t$ under the sparsity constraint, and the third term enforces that the updated weight generation hypernetwork maintains the previously learned object representations. The value of the negative scalar coefficients $\alpha, \beta$ will be detailed in the experiments.

**Backward redundancy removal.** The last but not the least component of the proposed object pursuit framework is to have a backward redundancy check. Since the weight generation hypernetwork is updated to $\psi^t = \psi^*$ with Eq. 4, there may now exist an embedding $\mu^*$ (computed using Eq. 3) that re-certifies object $o_t$ as an object falls on the manifold spanned by $z^{t-1}$ under $\psi^t$. If this is true, we set $z^t = z^{t-1}$, otherwise, $z^*$ is added to the list of base object representations since object $o_t$ is now confirmed as a base object. In some rare cases, object $o_t$ might be hard to learn, e.g., $z^*$ may not satisfy the criterion described in Eq. 1 under the current hypernetwork $\psi^t$. In this case, we simply toss away this object so that it can be better learned in the future as the pursuit process evolves. The proposed object pursuit framework is also summarized in Algorithm. 1.

## 4 EXPERIMENTS

We target the learning scenario where a scene consists of multiple objects, each of them can be discovered and manipulated through interactions. The objects are learned one by one in a continuous manner but with unknown orders. There are two main aspects of the whole pipeline, i.e., data collection by sampling the marginals of individual objects and construction of the object-centric representations with *Object Pursuit*. We focus on continuous object-centric representation learning, and thus orient our study on the behavior and characteristics of the proposed object pursuit algorithm. We also perform experiments on one-shot and few-shot learning, and show the potential of the learned object-centric representations in effectively reducing supervisions for object detection. Next, we brief our data collection process.

### 4.1 SETUP

**Data collection.** To learn diverse objects from variant positions and viewing angles, we collect synthetic data within the **iThor** environment ((Kolve et al., 2017)), which provides a set of interactive objects and scenes, as well as accurate modeling of the physics. We collect data of 138 different objects to generate their images and masks. The 138 objects are divided into 52 pretraining objects, 60 train objects for the pursuit process, and 25 test unseen objects. To focus on the representation learning part, we abstract the interaction policy, and the data collection procedure of a single object can be summarized as follows: 1) Randomly set the positions of all the objects in the scene. 2) Calculate all available camera positions and viewing angles from which the target object (to be learned) is visible so that the sampling is effective. The camera position, yaw angle, and pitch angle change within the range of 0.4 (grid size), $4°$ and $30°$ respectively. 3) For each camera position and viewing angle, we collect a $572 \times 572$ RGB image and a binary mask of the target object. 4) Repeat (1-3) for all objects in the stream. Please check Fig. 2 for the sampled data.

Table 1: Re-identification: recall and precision on seen objects.

| $\tau$ | 0.5 | 0.6 | 0.7 | 0.8 |
|---|---|---|---|---|
| recall | 1.0 | 1.0 | 1.0 | 1.0 |
| precision | 1.0 | 1.0 | 1.0 | 1.0 |

Table 2: Re-identification: rate of unseen objects been identified along the course of the pursuit process.

| $\tau$ | No. of trained objects | | | | | | |
|---|---|---|---|---|---|---|---|
| | 8 | 16 | 24 | 32 | 40 | 48 | 56 |
| 0.5 | 0.40 | 0.52 | 0.56 | 0.60 | 0.64 | 0.64 | 0.72 |
| 0.6 | 0.08 | 0.20 | 0.28 | 0.44 | 0.56 | 0.60 | 0.60 |
| 0.7 | 0.16 | 0.28 | 0.32 | 0.40 | 0.40 | 0.48 | 0.44 |
| 0.8 | 0.00 | 0.08 | 0.16 | 0.24 | 0.28 | 0.28 | 0.28 |

**Network implementation** In our experiment, we use Deeplab v3+ (Chen et al., 2018a) as the segmentation network $\phi$, which consists of 3 parts: a backbone to encode features at different levels, an aspp module, and a decoder to predict the segmentation probability per pixel. We use resnet18 as the backbone (encoder), whose weights are fixed both in the pretraining and the pursuit process. The weights of the aspp module and the decoder are generated by the convolutional hypernetwork $\psi$. For each convolution layer in the aspp module and the decoder, $\psi$ takes object representation $\mathbf{z}$ as input, and predict weights of the convolution kernel using an upsampling convolution block. The input representation $\mathbf{z}$ first expanded to a 1024-dim vector by a linear mapping and resized to a $1 \times 1 \times 32 \times 32$ tensor. After going through several upsampling blocks, each of which consists of an upsampling followed by a convolution and a leaky Relu, the $1 \times 1 \times 32 \times 32$ tensor turns into the output kernel weight. For other network weights like 'running_mean' and 'running_var' in batch normalization, the hypernetwork linearly maps representation $\mathbf{z}$ to generate them.

**Training details.** For the similarity measure $\Delta$, we use the dice score proposed in (Milletari et al.). In addition to $\Delta$, we find that it will be beneficial to add an extra binary cross-entropy term when learning base object representations using Eq. 4. Note, to deal with imbalanced foreground and background sizes, we also put a weighting on the entropy terms that correspond to the object so the learning can be more efficient. The sparsity constraint $\alpha$ is set to $-0.2, -0.1$ for Eq. 3 and Eq. 4 respectively, and $\beta = -0.04$ for all our experiments. To improve the convergence, we also warm up the hypernetwork using the pretraining objects. During pretraining, each mini-batch contains training data from one object, and we randomly choose which object to use in the next batch. In backpropagation, we update the hypernetwork $\psi$ and representation z for each object. When the pretraining is done, we perform a redundancy check to get rid of the objects that can be represented by others. For simplicity, this check is performed in sequential order, and we are left with a set of base object representations to carry out the following studies.

## 4.2 ON THE REPRESENTATION QUALITY MEASURE

The learning dynamics and the output of Algorithm. 1, i.e., the lists of base object representations $\mathbf{z}$ and the learned objects $\boldsymbol{\mu}$, together with the weight generation hypernetwork $\psi$, are primarily affected by the representation quality measure $\tau$ introduced in Eq. 1. For example, $\tau$ controls whether an object will be claimed as seen, and it also determines whether or not an object falls on the manifold spanned by the current base object representations. We study each of them in the following.

### 4.2.1 RE-IDENTIFICATION

As described in Eq. 2, when an object is discovered, it will be first checked against the learned objects and re-identified if the maximum expected similarity passes $\tau$. To examine how the quality measure $\tau$ influences the re-identification process, we run multiple object pursuit processes with different $\tau$'s. All runs are performed with the same training object order so that the only variant is the value of $\tau$. For evaluation, we preserve a separate set of 25 objects (unseen test objects) that never appear during training. Note, among these unseen test objects, there are also objects that are similar to the training ones. And we use 27 objects (seen test objects) from the warp-up joint training described above to check the re-identification accuracy.

First, we check how $\tau$ affects the re-identification for seen objects. As reported in Tab. 1, if an object is learned and added to the object list $\boldsymbol{\mu}$, it will be claimed as seen by Eq. 2, and the re-identification accuracy is always one. This is true for $\tau$ varying between 0.5 and 0.8, which demonstrates the robustness of the re-identification process against $\tau$ for objects learned.

Second, we check the behavior of the re-identification component for unseen objects under different $\tau$'s along the pursuit process. In Tab. 2, we can observe that as more and more objects are learned during the pursuit, the unseen objects that are claimed as seen from the re-identification process also increase. This observation is consistent across different $\tau$'s. Furthermore, the rate of unseen objects identified as seen converges at the end of the pursuit process, but at different levels for different $\tau$'s, i.e., the converged rate is lower for larger $\tau$. It may seem incorrect if an unseen object is claimed as seen by the re-identification component. However, if we examine the unseen objects (see Fig. 3), we can see that it is quite natural for these unseen objects to be labeled as seen, because they are similar to one or multiple objects in the object list $\boldsymbol{\mu}$. This is indeed a desired characteristic since representing

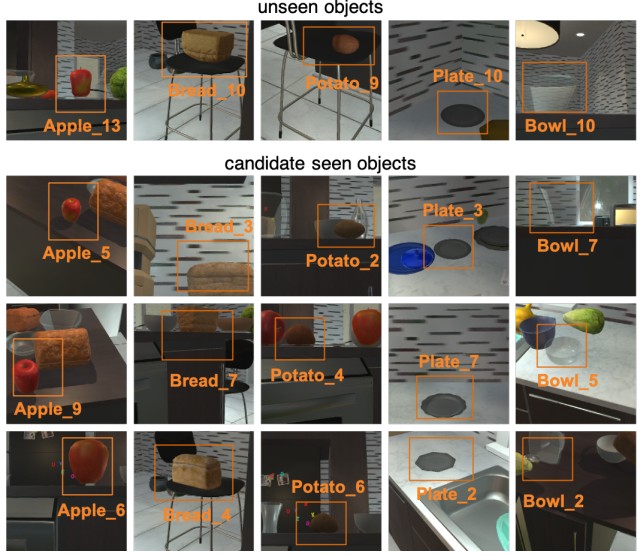

Figure 3: Unseen objects re-identified as learned. 1st row: unseen objects, 2nd to 4th row: similar objects from the learned object list. Bounding boxes highlight the objects with embedded text indicating the instance identity.

or learning an object that is similar to existing ones may not be informative. Moreover, one can adjust $\tau$ to tune the similarity level. For example, if one insists on learning an object similar to previously seen objects, increasing the value of $\tau$ should work as evidenced by the converged rates for $\tau$'s in Tab. 2.

In a nutshell, the representation quality measure $\tau$ has little effect on the re-identification recall and accuracy for learned objects. Yet, it controls the granularity of the learned representations by modulating the rate of unseen objects that would be identified as learned ones.

### 4.2.2 SUCCINCTNESS AND EXPRESSIVENESS

We want to study how the representation quality measure $\tau$ affects the overall learning dynamics in terms of the succinctness and expressiveness of the learned base object representations. By checking Eq. 2, Eq 3 and also the previous experiment, we conjecture that if $\tau$ is small, objects similar to the learned ones will be more easily identified as seen and certified as on the manifold. If so, the number of objects that will be used for learning the base representations may also be small, thus increasing the succinctness of the final representations. Conversely, when $\tau$ increases, we would expect that more objects will contribute to the base representations, thus increasing the expressiveness. We like to check if the observations align with our conjecture and how such behav-

Table 3: Pursuit dynamics by varying $\tau$. Please see the enclosing description for the meaning of the metrics and corresponding analysis.

| $\tau$ | 0.5 | 0.6 | 0.7 | 0.8 |
|---|---|---|---|---|
| $|\mathbf{z}|/N$ | 0.34 | 0.42 | 0.42 | 0.40 |
| $|\boldsymbol{\mu}|/N$ | 0.46 | 0.58 | 0.46 | 0.40 |
| $\mathcal{R}_e$ | 0.19 | 0.21 | 0.00 | 0.00 |
| $\mathcal{R}_f$ | 0.08 | 0.18 | 0.42 | 0.54 |
| $\mathcal{A}_\mu$ | 0.75 | 0.77 | 0.83 | 0.86 |

ior affects the quality of the learned bases. To facilitate the analysis, we propose to check the following quantities: 1) $|\mathbf{z}|/N$, which is the portion of objects that contribute to base representations; 2) $|\boldsymbol{\mu}|/N$, which is the portion of learnable objects that are added to the object list $\boldsymbol{\mu}$; 3) $\mathcal{R}_e$, rate of objects that are confirmed unseen but can be expressed by the base object representations; 4) $\mathcal{R}_f$, rate of objects to be learned as base representations, which are later considered as redundant or unqualified; 5) $\mathcal{A}_\mu$, segmentation accuracy on learned objects.

We report the above metrics across different $\tau$'s in Tab. 3. As expected, a larger $\tau$ generally encourages more objects to be learned as base representations. For example, the number of base objects learned is much larger when $\tau$ increases from 0.5 to 0.6 (first row). This is also evidenced by the third row of Tab. 3, which shows that the probability of an unseen object to be expressed by base rep-

resentations will decrease as $\tau$ increases, creating more attempts to learn objects as bases. However, the number of learnable objects, i.e., a base object or an object that falls on the manifold spanned by the bases, attains the maximum at a medium value $\tau = 0.6$ (second row). The underlying reason is two-fold: First, a very small $\tau$ means that many objects will be identified as seen and thus discarded to save computation; Second, a very large $\tau$ can make the qualification of an object representation extremely difficult such that it will be put aside for future learning. The latter is also supported by the metric $\mathcal{R}_f$ shown in the fourth row, i.e., the probability that an object will be considered redundant or unqualified after learning as a base object will increase as $\tau$ becomes large. Lastly, when checking the quality of the base representations in expressing a common set of learned objects, we can see that the segmentation accuracy correlates with $\tau$ in a positive manner (fifth row).

In general, $\tau$ directly impacts the quality of the base representations for learned objects, but its effect on the number of base representations produced by the pursuit procedure is not monotone. Within a moderate range, we can increase $\tau$ to encourage learning more base representations, however, we may not want $\tau$ to be too large that only a few objects are qualified as base representations.

### 4.2.3 LABEL EFFICIENCY

Besides the training dynamics, we evaluate the usefulness of the learned object base representations in terms of how it facilitates learning the representation of a new object with only a few annotations. For comparison, we also perform learning of the object representations over the entire representation space. Training is similar to Eq. 3. The quality of the learned object representations is measured by their segmentation accuracy on test data.

Table 4: N-shot learning the representation of a new object. Training is performed by searching the optimal representation either on the manifold spanned by the base objects, or over the entire representation space. Segmentation accuracy on the test set is reported for bases and hypernetworks learned at different $\tau$'s.

| n | over base object representations | | | | full representation space | | | |
|---|---|---|---|---|---|---|---|---|
| | 0.5 | 0.6 | 0.7 | 0.8 | 0.5 | 0.6 | 0.7 | 0.8 |
| 1 | 0.377 | 0.416 | 0.454 | 0.446 | 0.225 | 0.264 | 0.288 | 0.289 |
| 5 | 0.595 | 0.606 | 0.634 | 0.614 | 0.461 | 0.475 | 0.468 | 0.453 |
| 10 | 0.622 | 0.647 | 0.677 | 0.649 | 0.542 | 0.526 | 0.524 | 0.520 |
| 2000 | 0.697 | 0.731 | 0.740 | 0.731 | 0.669 | 0.698 | 0.702 | 0.718 |

As reported in Tab. 4, the quality of the few-shot learned representations increases as $\tau$ gets large, which aligns with our observation in the previous section that the expressiveness of the learned object base representations highly correlates with $\tau$. However, note that there is a slight drop in performance when $\tau$ increases from 0.7 to 0.8 (fourth and fifth column). The reason is that as $\tau$ gets really large, it also becomes much easier to omit objects that can not pass the quality test. As a result, the hypernetwork, which translates the representation to network weights, also gets less trained. Thus, when tested on new objects, the performance may not match that of the trained objects for the same set of base representations, suggesting again that a moderate $\tau$ is needed to balance between the succinctness and generalization of the learned base representations.

The above observation does not hold for the representations learned over the full space. Moreover, when comparing the performance within the low data regime, we can see that those object representations found on the manifold outperform those found in the entire space by a large margin. For example, the new object representations found with the learned bases under $\tau = 0.7$ outperform their counterparts by $57.6\%, 35.5\%, 29.2\%$ for the 1-shot, 5-shot, and 10-shot settings, respectively. This demonstrates the potential of using the learned base representations to help reduce the supervision needed to learn a new object. Also, it confirms that the learned base representations are meaningful since the manifold spanned by them provides a good regularity for learning unseen objects.

### 4.3 ORDER OF TRAINING OBJECTS

The proposed object pursuit algorithm learns object representations in a stream, so we also check how the learning dynamics vary when the order of training objects changes. We fix $\tau$ to 0.6 and run ten pursuit processes with random training object order. We reported the mean and standard deviation of the metrics proposed in Tab. 3. As observed in Tab. 5, the pursuit process is robust to the training object order.

Table 5: Pursuit dynamics under random training object order.

| | $|\mathbf{z}|/N$ | $|\boldsymbol{\mu}|/N$ | $\mathcal{R}_e$ | $\mathcal{R}_f$ | $\mathcal{A}_\mu$ |
|---|---|---|---|---|---|
| mean | 0.43 | 0.50 | 0.10 | 0.15 | 0.76 |
| std-dev | 0.02 | 0.03 | 0.04 | 0.04 | 0.01 |

### 4.4 Forgetting Prevention

In this section, we want to check if the forgetting prevention term in Eq. 4 is effective and how it affects the pursuit dynamics. We run pursuit processes with different values of the coefficient $\beta$, where the quality measure $\tau$ and training object order is fixed. Segmentation accuracy $\mathcal{A}_\mu$ and forgetting rate $\gamma_f$ (i.e., how many objects falls under the quality measure after the process is finished) in Tab. 6 demonstrate the effectiveness of the forgetting prevention term: when $|\beta|$ decreases, the segmentation accuracy drops, and the forgetting rate increases; when $|\beta|$ vanishes, the forgetting rate reaches 97%, which means that the hypernetwork almost forgets all the object representations it previously learned. We can also observe that both $|\mathbf{z}|/N$ and $|\boldsymbol{\mu}|/N$ increase

Table 6: Pursuit dynamics under different forgetting prevention constraints.

| $|\beta|$ | 0.0 | 0.02 | 0.04 | 0.1 |
|---|---|---|---|---|
| $|\mathbf{z}|/N$ | 0.61 | 0.46 | 0.42 | 0.39 |
| $|\boldsymbol{\mu}|/N$ | 0.88 | 0.61 | 0.58 | 0.54 |
| $\mathcal{R}_e$ | 0.13 | 0.14 | 0.21 | 0.21 |
| $\mathcal{R}_f$ | 0.19 | 0.14 | 0.18 | 0.21 |
| $\mathcal{A}_\mu$ | 0.02 | 0.67 | 0.71 | 0.72 |
| $\gamma_f$ | 0.97 | 0.04 | 0.02 | 0.02 |

when $|\beta|$ decreases. This is due to the fact that when the hypernetwork forgets what are learned, any incoming object will be unlikely to be considered as seen, nor to be expressed by current bases. So the hypernetwork tends to learn them as new base objects, which causes $|\mathbf{z}|/N$ to increase. This is also evidenced by the drop in $\mathcal{R}_e$, which is rate of new objects that are certified as on the object manifold. Furthermore, without the constraint of the forgetting prevention term, it is more likely to get higher accuracy in learning a new object, which decreases the number of unqualified objects. Since the number of redundant objects and unqualified objects both drop when $|\beta|$ decreases, $|\boldsymbol{\mu}|/N$ increases. Thus, in order to reduce computational cost and enforce learning meaningful representations, one would like to apply a relatively large $|\beta|$ during the pursuit process.

We can also observe that as $|\beta|$ changes from 0.02 to 0.1, $\mathcal{R}_f$ increases monotonically, this is because the forgetting prevention constraint affects the quality of the learned representations, since less freedom is available when $|\beta|$ is extremely large. Consequently, fewer objects will be qualified with a good representation measured by $\tau$. On the other hand, $\mathcal{R}_f$ is also high when $beta$ is set to 0. The reason is that when learning a new object without the constraint of the forgetting prevention term, the hypernetwork tends to overfit, thus making it easier for this new object to be considered as redundant, i.e., it can be expressed by the existing base representations, even though the learned representation will be forgotten by the network right after the current learning episode.

### 4.5 More Results

In the appendix, we also provide ablation studies on how the sparsity constraints in Eq. 3 and Eq. 4 affect the learning dynamics and the quality of the learned representations. By examining the most relevant base objects for a novel object that can be expressed by the base representations (Fig. 4), we can qualitatively see that high-level concepts are learned within the representation space as objects that share similar geometry or appearance will be more correlated than others. For curiosity, we also test the usefulness of the base object representations on real-world video objects. As demonstrated in Fig. 7, the learned base representations can capture well the representations of real-world objects with a single learning example even if they are trained on synthetic data.

## 5 Conclusion

We demonstrate that the proposed *object pursuit* framework can be used for continuously learning object-centric representations from data collected by manipulating a single object. The key designs, e.g., object re-identification, forgetting prevention, and redundancy check, all contribute to the quality of the learned base object representations. We also show the potential of using the learned object-centric representations for tasks at a low-annotation regime. Especially, the learned object manifold provides a meaningful and effective prior on objects, which can facilitate downstream tasks that require object-level reasoning. As inspired by an initial attempt on the real-world data (Fig. 7), we would also like to check the proposed object pursuit algorithm in the real world. For example, we can train an autonomous agent to collect data from the natural 3D environment with a more efficient interaction policy, and then test the learned object representations on real-world compositional visual reasoning tasks. These are in our future research agenda.

**Acknowledgement**    Toyota Research Institute provided funds to support this work. It is also supported by a Vannevar Bush Faculty Fellowship and a grant from the Stanford HAI Institute.

**Ethics Statement.**    The proposed *object pursuit* framework aims at learning object representations for object-centric visual reasoning tasks. Currently, the experiments are performed in simulation, which is publicly available and comes with a proper license. However, how to use the learned representations could be an issue, and we will explicitly state the guideline on how to use our code and trained models ethically. In our future research, when data collection in the real world is involved, we will consult the university ethics review committee for advice. However, in the current form, we do not observe any significant concerns.

**Reproducibility Statement.**    Our code, training data, and learned models are made publicly available. We add detailed comments in the code so that the implementation can be easily understood. For a preview of the implementation, please refer to the attached code in the supplementary materials.

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

# A APPENDIX

## A.1 ALGORITHM

Here is the Object Pursuit algorithm we describe in the method section. In this algorithm, we first check if the object is seen according to Eq. 2. If the object is seen, we directly move on to the next object, otherwise, we start to check if the object can be represented by current bases according to Eq. 3. If it can be expressed, we add this object to the object list $\boldsymbol{\mu}$ and move on to the next object, otherwise, we need to train for a new $z$ and update the hypernetwork using loss function Eq. 4. After training, we start a second redundancy check, with the same criterion Eq. 3 in the first check. If the object can be represented by bases, we just add it to the object list $\boldsymbol{\mu}$, otherwise, we consider it a new base and add it to both base list $\mathbf{z}$ and object list $\boldsymbol{\mu}$. The algorithm runs in a loop to simulate that objects are continually showing up and learned by our system.

---

**Algorithm 1:** Object Pursuit

---

**Result:** A set of object representations $\boldsymbol{\mu}^T = \{\mu_i\}_{i=1}^N$, $\mathbf{z}^T = \{z_i\}_{i=1}^B$, and the hypernet $\psi^T$,
        with $T \geq N \geq B$
initialization: $\mathbf{z}^0 = \boldsymbol{\mu}^0 = \emptyset$, $\psi^0$ is randomly initialized;
**while** $t \leq T$ **do**
     Sample $\mathcal{D}^t = \{(\mathbf{x}_j, \mathbf{y}_j^t)\} \sim \mathbf{p}(\mathbf{x}, \mathbf{y}^t)$;
     Check if object $o_t$ is in $\boldsymbol{\mu}^{t-1}$ (parameter $\tau$);
     **if** *YES* **then**
         $\boldsymbol{\mu}^t = \boldsymbol{\mu}^{t-1}$;
         $\mathbf{z}^t = \mathbf{z}^{t-1}$;
         $\psi^t = \psi^{t-1}$;
     **else**
         Check if object $o_t$ can be represented using $\mathbf{z}^{t-1}$ (parameter $\tau$);
         **if** *YES* **then**
             $\boldsymbol{\mu}^t = \boldsymbol{\mu}^{t-1} \bigcup \mu_{o_t}$;
             $\mathbf{z}^t = \mathbf{z}^{t-1}$;
             $\psi^t = \psi^{t-1}$;
         **else**
             Training for $z_{o_t}$ and $\psi'$ under the constraint of all objects in $\boldsymbol{\mu}^{t-1}$;
             $\psi^t = \psi'$;
             Check if $z_{o_t}$ can now be approximated by $\mathbf{z}^{t-1}$;
             **if** *YES* **then**
                 $\boldsymbol{\mu}^t = \boldsymbol{\mu}^{t-1} \bigcup \mu_{o_t}$;
                 $\mathbf{z}^t = \mathbf{z}^{t-1}$;
             **else**
                 $\boldsymbol{\mu}^t = \boldsymbol{\mu}^{t-1} \bigcup [0, 0, ..., 0, 1]$;
                 $\mathbf{z}^t = \mathbf{z}^{t-1} \bigcup z_{o_t}$;
             **end**
         **end**
     **end**
**end**

---

## A.2 ABLATION ON THE SPARSITY CONSTRAINTS

### 1. **L-1 norm coefficient on $\mu$**

In this ablation study, we aim to understand whether L-1 norm on combination coefficient $\mu$ in Eq. 3 affects an object falls on the manifold or not. From Tab. 7 we can see that when L-1 norm coefficient $|\alpha|$ change from 0.0 to 0.2 (absolute value), $\mathcal{R}_e$ increases, which means more object can be expressed by current bases, causing the number of base objects ($|\mathbf{z}|/N$) to decrease. It shows that if we properly increase the constraint on $\mu$ and promote the sparsity of $\mu$, it may be easier to find a $\mu$ that can express the object. However, if $|\alpha|$ exceeds a certain limit, e.g. 0.5, too much constraint added to $\mu$ (which makes $\mu$ harder to change), finding

Table 7: Pursuit dynamics under different L-1 norm coefficients on $\mu$

| $|\alpha|$ | 0.0 | 0.1 | 0.2 | 0.5 |
|---|---|---|---|---|
| $|\mathbf{z}|/N$ | 0.45 | 0.42 | 0.42 | 0.40 |
| $|\boldsymbol{\mu}|/N$ | 0.57 | 0.55 | 0.58 | 0.51 |
| $\mathcal{R}_e$ | 0.11 | 0.19 | 0.21 | 0.08 |
| $\mathcal{R}_f$ | 0.21 | 0.15 | 0.17 | 0.25 |
| $\mathcal{A}_\mu$ | 0.74 | 0.75 | 0.73 | 0.77 |

a $\mu$ to express an object becomes difficult, thus the number of objects that can be expressed ($\mathcal{R}_e$) decreases, as shown in Tab. 7.

We can also see that $\mathcal{R}_f$ first drops then increases when $|\alpha|$ gets bigger. Our reasoning is that, even though more objects can not be expressed by the bases at the first redundancy check and have to be learned as new representations when $|\alpha|$ increases, the chances that they are unqualified or redundant after training also increase, making the base number continually decreases.

### 2. **L-1 norm coefficient on $\mathbf{z}$**

In this ablation study, we focus on how L-1 norm on $\mathbf{z}$ affects object pursuit. We change the coefficient $\alpha$ and evaluate the pursuit process. As Tab. 8 shows, when $|\alpha|$ increases, the number of objects that can be expressed by bases ($\mathcal{R}_e$) also increases, causing the base number ($|\mathbf{z}|/N$) to decrease. This is because regularization on $\mathbf{z}$ prevents the object representations from getting too far from each other, thus making $\mathbf{z}$ distribute more uniformly. Since $\mathbf{z}$ is well distributed, it may be easier to find a coefficient $\mu$ that can express an object. We also find it in our experiment that when $|\alpha|$ gets bigger, more objects will be considered

Table 8: Pursuit dynamics under different L-1 norm coefficient on $\mathbf{z}$

| $|\alpha|$ | 0.0 | 0.1 | 0.2 | 0.5 |
|---|---|---|---|---|
| $|\mathbf{z}|/N$ | 0.46 | 0.42 | 0.42 | 0.39 |
| $|\boldsymbol{\mu}|/N$ | 0.56 | 0.58 | 0.56 | 0.55 |
| $\mathcal{R}_e$ | 0.12 | 0.21 | 0.21 | 0.24 |
| $\mathcal{R}_f$ | 0.14 | 0.18 | 0.12 | 0.19 |
| $\mathcal{A}_\mu$ | 0.74 | 0.72 | 0.74 | 0.72 |

as unqualified due to their low training accuracy, especially when $|\alpha| = 0.5$. It shows that more constraints on $\mathbf{z}$ may cause it harder to find a proper $\mathbf{z}$ to represent an object during training, thus decrease the accuracy. It also explains why the number of learnable objects ($|\boldsymbol{\mu}|/N$) decreases when $|\alpha|$ change from 0.1 to 0.5.

## A.3 UNDERSTAND THE BASE REPRESENTATIONS

Fig. 4 shows some visualization results of unseen objects and the corresponding active base objects. The results are from the pursuit process. When an unseen object is detected, and if it can be expressed by the current bases, we then find the combination coefficient $\mu$ through Eq. 3. In Fig. 4, we show three examples that the unseen objects can be expressed by base objects, and the combination coefficient values are shown below the corresponding bases. We show the top 5 bases that have the highest coefficient value among all the bases.

In the first example (the 1st row), the base object 'DishSponge_1' has the highest coefficient value in expressing a green cup (Cup_3). We conjecture that 'Cup_3' and 'DishSponge_1' share a similar color, and green objects are rare in this set of bases, causing the DishSponge's coefficient to be the highest one. 'Dumbell_1' may share a similar shape with 'Cup_3', since they are both thin in the middle and thick in the end. The second example (the 2nd row) shows that if there is a base object (Bowl_7) that is evidently more similar to the target object (Bowl_10) than other bases, its coefficient value may be the highest. Here Bowl_7 and Bowl_10 are similar in both color and shape, but they are different in shape at the bottom of the bowl. In the third example (the 3rd row), the black cup (Cup_2) and the black pot (Pot_1) share the same color, while the black cup (Cup_2) and the glass cup (Cup_1) share a similar shape.

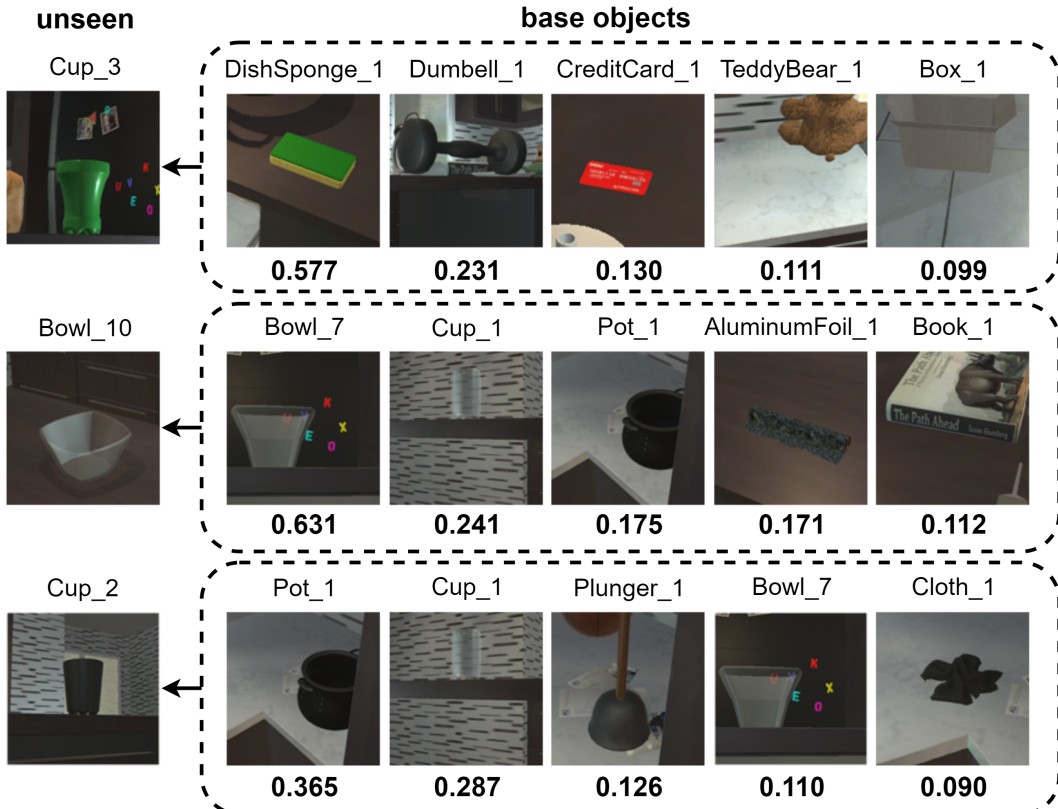

Figure 4: Unseen objects expressed by base objects. Unseen objects in the 1st column (Cup_3, Bowl_10, Cup_2) are represented by base objects in the 2nd to 6th columns with combination coefficients shown below the object. Higher coefficient means greater weight or importance in the combination.

These results show that the object representation space learned through object pursuit characterizes some high-level concepts (e.g. color, shape) that help better understanding the objects. An unseen object tends to be represented by a base object with similar color or shape. Although in some cases, understanding the base representations is not as easy as the examples shown in Fig. 4, this regular pattern is still evident in most cases.

### A.4    More Visual Results on Synthetic Data

Fig. 5 shows more segmentation results from the experiments. In these examples (the 1st row and the 2nd row), 'bowl' and 'kettle' are two unseen objects which are not in the object list $\mu$ and the base list $\mathbf{z}$. To represent an unseen object, we fix the hypernetwork and try to find a combination coefficient $\mu^*$ to express the target object with the bases, according to Eq. 3. The unseen object, which resides on the manifold (the object space) with latent code $\mathbf{z}$ can be segmented by the segmentation network generated from its representation with the hypernetwork. Note, we test the segmentation network on the validation dataset.

Even if the unseen object is expressed by base objects which may share some similarity with it, the unseen object can still be segmented accurately from various backgrounds in different positions and angles, as Fig. 5 shows. It verifies that the learned representations, together with the hypernetwork, preserve discriminative object-centric knowledge.

### A.5    Real-World Data Collection and Learning

We propose that our framework *object pursuit* can be used with autonomous agents that explore the environment and interact with objects. Through interactions, an agent could learn object-centric representations, and this is an unsupervised setting since *no human annotations are required* when the agent interacts with objects in the physical scene.

novel object 'bowl'

novel object 'kettle'

Figure 5: More segmentation results. The objects showed above are unseen objects that are expressed by base objects.

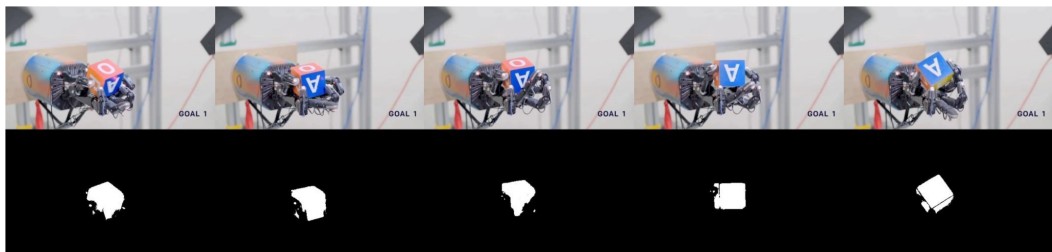

Figure 6: Demonstration of obtaining instance masks of manipulated objects through interaction. 1st row: images of a robotic arm manipulating a cube. 2nd row: instance masks obtained from interaction by motion segmentation and edge refinement.

Currently, we perform the major experiments on synthetic data due to the lack of a robot to perform the data collection in the real world. Moreover, we like to have a proof of concept before carrying out real-world experiments, which could involve an enormous amount of research funds and effort, which is out of the scope of our current work. However, we are confident that transitioning from the synthetic environment to the real world is highly practical as all the technical components that need to be used in the real world are ready.

In order **to evaluate how practical it is** for a robot to perform the pursuit process in the real world, we need to look into two aspects. First, how possible it is to obtain instance masks of the manipulated object. Second, how efficient the learning can be given the objects that need to be learned. Next, we demonstrate how it is possible to get the instance masks of a real-world object through interaction. And we discuss the second aspect in the following section.

### A.5.1 COLLECT DATA IN THE REAL-WORLD VIA INTERACTION

In our setting, a robot manipulates only one object at a time and learns its representation. Two cues can be used to obtain the instance mask of the object being manipulated. **Robotic arm localization and motion.** It is easy to know where the robotic arm is within the view through calibration. Also, motion segmentation is a well-studied topic on real-world objects in the literature (Charig Yang et al., 2021; Yang et al., 2019). A possible pipeline is to first apply motion segmentation on the images, and this should give the masks of both the robotic arm and the object as they are both moving. To remove the portion of the robotic arm, the agent can treat its arm as the first object to learn, so that it knows how to segment its arm in the future. By subtracting the arm, it now has the mask of the manipulated object.

Fig. 6 shows an example of getting instance masks from a video of object manipulation without supervision. In this example, the background in the video is static (but the motion segmentation algorithm we employed can also work when there is ego-motion), and we first segment the cube together with the robot hand using motion segmentation (Charig Yang et al., 2021). Then we remove the robot hand from the segmentation mask, using "densecrf" (Krähenbühl & Koltun, 2011) as post-processing. As mentioned, there are other substitutes to exclude the hands, such as using a

specifically trained segmentation model of the robotic arm. The segmentation process can run at 5 frames per second. And the results show the practicality of obtaining instance masks during manipulating objects in the real world.

### A.5.2 REAL-WORLD OBJECT PURSUIT

To evaluate the efficiency and robustness of learning on real-world images or videos, we conduct experiments on two large-scale real-world datasets, i.e., Youtube VOS (Xu et al., 2018) and CO3D (Reizenstein et al., 2021).

**YouTube-VOS** We train and evaluate our framework on the Youtube-VOS dataset, which contains 65 categories. Each category may appear in multiple video sequences, and there may be multiple instances of the same category in a video sequence. We traverse all the categories, and for each category, we sample one video sequence from all sequences containing a single instance of this category to serve as the object we pursuit (Note this is to stimulate the interaction that should happen in the real world). The number of frames in a video sequence typically ranges from 20 - 36. For each video sequence, we repeat the frames many times so that the effective dataset size is around 500 samples. For each frame, we apply horizontal flip augmentation and random crop augmentation. Specifically, we crop each frame with the same ratio along the x and y-axis, which is a random number between 0.6 and 1. Finally, we resize the frame to 320x180 (The origin size is 1280x720).

**CO3D** We also test our framework on CO3D. Since processing the original dataset is too time-consuming (18,619 objects of 50 MS-COCO categories), we randomly select 285 objects from 8 object classes: apple, banana, backpack, baseball bat, bench, bicycle, book, and bottle as training objects. The data of each object contains a video sequence that shows different views of the object. We randomly crop each frame to a square image, then apply horizontal flip augmentation. After augmentation, the effective dataset size is about 200 samples per object. Each frame is at the resolution 256x256.

**Other training details.** For both datasets, we initialize the hypernet with the model pretrained on synthetic data, with no initial bases and objects. We set different quality measures $\tau$ (0.5, 0.6, 0.7, and 0.8) to evaluate our framework on real-world data. Other settings are the same as the synthetic data experiment.

Table 9: Pursuit dynamics on YouTube-VOS dataset by varying $\tau$.

| $\tau$ | 0.5 | 0.6 | 0.7 | 0.8 |
|---|---|---|---|---|
| $|\mathbf{z}|/N$ | 0.35 | 0.38 | 0.38 | 0.40 |
| $|\boldsymbol{\mu}|/N$ | 0.48 | 0.49 | 0.49 | 0.51 |
| $\mathcal{R}_e$ | 0.20 | 0.16 | 0.14 | 0.10 |
| $\mathcal{R}_f$ | 0.30 | 0.32 | 0.52 | 0.58 |
| $\mathcal{A}_\mu$ | 0.72 | 0.76 | 0.81 | 0.84 |

Table 10: Pursuit dynamics on CO3D dataset by varying $\tau$.

| $\tau$ | 0.5 | 0.6 | 0.7 | 0.8 |
|---|---|---|---|---|
| $|\mathbf{z}|/N$ | 0.15 | 0.21 | 0.22 | 0.28 |
| $|\boldsymbol{\mu}|/N$ | 0.19 | 0.29 | 0.29 | 0.37 |
| $\mathcal{R}_e$ | 0.14 | 0.24 | 0.16 | 0.11 |
| $\mathcal{R}_f$ | 0.37 | 0.38 | 0.55 | 0.58 |
| $\mathcal{A}_\mu$ | 0.69 | 0.74 | 0.82 | 0.87 |

**Pursuit dynamics.** Tab. 9 and Tab. 10 show pursuit dynamics under different $\tau$ on YouTube-VOS and CO3D datasets, respectively. The real-world data experiments share similar patterns with the synthetic data experiments: when $\tau$ gets bigger, $\mathcal{A}_\mu$ (average segmentation accuracy) and $\mathcal{R}_f$ (proportion of unqualified and redundant objects) gets bigger, while $\mathcal{R}_e$ (proportion of expressed objects) first increases then decreases. A higher $\tau$ excludes objects with poor training accuracy, thus improving the overall segmentation performance. On the other hand, a higher $\tau$ could make an object easier to be considered as unqualified, which would be skipped and would not be learned by our system. The number of bases and qualified objects both increase with $\tau$, because fewer objects would be claimed as seen when $\tau$ increases, increasing the number of learned objects. The similar patterns between real-data and synthetic data show that the conclusions we made from Tab. 3 can generalize to the real-world domain and transfer between different datasets.

**Learning efficiency.** The running time of our framework depends on the threshold $\tau$. Generally, a smaller $\tau$ leads to a shorter running time since objects are easier to be expressed by bases or recognized as seen objects, which will save the time of learning base object representations. In the synthetic data experiments, $\tau = 0.5$ could finish running 67 objects in about 10 hours, while $\tau = 0.8$

needs 1.5 days. In the real-world experiments on youtubeVOS and CO3D datasets, our framework can learn around 80 objects per day under $\tau = 0.5$ and 40 objects per day under $\tau = 0.8$.

**Conclusion on real-world practicality.** With these experiments, we can conclude that the observations we make in the synthetic environment transfer to real-world data. Moreover, **two key factors** that are directly related to how practical it is to run *object pursuit* in the real world are **checked to be positive.** First, collecting data with object instance masks through interaction in the real world is practical, as verified by the effectiveness of the proposed pipeline in the previous section. Second, the pursuit framework is robust on the real-world data, and the efficiency is good enough to perform learning in the real world. For example, in a house with hundreds of objects, the training can be finished within three weeks and requires no human supervision, which is far more time-consuming and complicated in practice.

### A.6 ADDITIONAL RESULTS ON RE-IDENTIFICATION

In section 4.2.1, we demonstrate the impact of the quality measure $\tau$ on seen objects and unseen objects separately during re-identification. To further elaborate on the impact of $\tau$, in this section, we report precision and recall on both seen objects and unseen objects (collectively) on the scale of all testing objects. For unseen objects, we define recall as the fraction of correctly identified unseen objects among all the unseen objects, and precision is defined as the fraction of correctly identified unseen objects among all the objects we identify as unseen. Same for seen objects.

Table 11: Re-identification: recall and precision of unseen objects (on all testing objects).

| $\tau$ | 0.5 | 0.6 | 0.7 | 0.8 |
|---|---|---|---|---|
| recall | 0.28 | 0.40 | 0.56 | 0.72 |
| precision | 1.0 | 1.0 | 1.0 | 1.0 |

Table 12: Re-identification: recall and precision of seen objects (on all testing objects).

| $\tau$ | 0.5 | 0.6 | 0.7 | 0.8 |
|---|---|---|---|---|
| recall | 1.0 | 1.0 | 1.0 | 1.0 |
| precision | 0.60 | 0.64 | 0.71 | 0.80 |

Tab. 11 and Tab. 12 report recall and precision on unseen objects and seen objects, taking all testing objects into consideration. We collect the number of objects our model identifies as seen or unseen from the re-identification experiment introduced in Section 4.2.1, then compute recall and precision. From these two tables, the precision of unseen objects and the recall of seen objects are 1.0 under all $\tau$, which shows that *a seen object can always be identified correctly*. This is *crucial to our framework*: if a seen object is not identified as seen, it would be problematic since we have to learn that object repeatedly, unlimitedly enlarging the object list $\boldsymbol{\mu}$. On the other hand, the recall of unseen objects and the precision of seen objects are lower than 1.0, showing that some unseen objects could be identified as seen. By visualizing the data (Fig. 3), we find this situation happens only when the unseen object has substantial similarities., e.g., in terms of color or shape, to a seen object. It has no significant impact on the training process, since these "seen but actually unseen" objects contribute very little information to the learned object representations. Therefore, our model *learns object representations in a way that prevents learning the same or highly similar objects*, improving efficiency.

We also find that the recall of unseen objects and the precision of seen objects get larger when $\tau$ gets bigger. It is because a larger $\tau$ increases the bar of determining an object as seen, according to Eq. 2. Therefore, a difference would make the model consider two as different objects. As $\tau$ gets bigger, the learned representation becomes more discriminative, at the cost of generalization, which is a characteristic that allows us to tune the system depending on the actual need.

### A.7 EVALUATION ON BASE AND NON-BASE OBJECTS

Tab. 3 shows pursuit dynamics, including segmentation accuracy and re-identification rate with different $\tau$ for base and non-base objects. We take all objects in the object list $\boldsymbol{\mu}$ into consideration. However, these objects are added to the object list in two different ways: some can be expressed by current bases (non-base), while others are trained and accepted as a new base (base). For the former, we simply find combination coefficients $\mu$ without updating the hypernet, optimizing the model in much smaller parameter space than the latter. This may cause different pursuit dynamics in these

two situations. This section analyzes the segmentation accuracy and re-identification rate to promote the understanding on this aspect.

As reported in Tab. 13, for both base and non-base objects, $\mathcal{A}_\mu$ shows the average segmentation accuracy, and $\mathcal{R}_{reid}$ shows the proportion of the objects that are correctly re-identified as seen objects. We report $\mathcal{A}_\mu$ and $\mathcal{R}_{reid}$ under different $\tau$. For $\tau = 0.8$, we found that all objects that appeared in the pursuit sequence could not be expressed by bases at first and would be learned as a new base, since the threshold $\tau$ is too high for an object to be considered as being expressed by bases. In this case, we could not compute

Table 13: Segmentation accuracy and re-identification rate on base and non-base objects.

| | $\tau$ | 0.5 | 0.6 | 0.7 | 0.8 |
|---|---|---|---|---|---|
| non-base | $\mathcal{A}_\mu$ | 0.65 | 0.69 | 0.73 | N/A |
| | $\mathcal{R}_{reid}$ | 1.0 | 1.0 | 1.0 | N/A |
| base | $\mathcal{A}_\mu$ | 0.78 | 0.82 | 0.84 | 0.86 |
| | $\mathcal{R}_{reid}$ | 1.0 | 1.0 | 1.0 | 1.0 |

$\mathcal{A}_\mu$ and $\mathcal{R}_{reid}$ for non-base objects. For other $\tau$, the re-identification rate $\mathcal{R}_{reid}$ stays stably at 1.0, showing that if a base or non-base object has been encountered, it could be re-identified correctly. Tab. 13 shows that $\mathcal{A}_\mu$ of both base and non-base objects increases when $\tau$ gets bigger, which shares a similar pattern with Tab. 3.

We also find that $\mathcal{A}_\mu$ on non-base objects could be lower than $\mathcal{A}_\mu$ on base objects. It indicates that updating the hypernet and training as a new base can perform better than simply combining bases due to a larger degree of freedom. This difference can be reduced by increasing the capacity of the hypernetwork. In conclusion, the re-identification performance is stable and accurate on both base and non-base objects, and the segmentation accuracy increases with $\tau$. Furthermore, the segmentation accuracy of base objects is generally higher than non-base objects.

## A.8 ONE-SHOT LEARNING ON REAL DATA

We perform one-shot learning on the DAVIS 2016 dataset (Perazzi et al., 2016), a video object segmentation dataset in the real scene. Under the one-shot learning scheme, we fix the hypernet and the bases, initialize the combination coefficient $\mu$ with only one

Table 14: Jaccard index on DAVIS evaluation set.

| Measure | ObjP. (Ours) | OSVOS | SEA | HVS | JMP |
|---|---|---|---|---|---|
| J Mean ↑ | **64.5** | 62.1 | 50.4 | 54.6 | 57.0 |
| J Recall ↑ | **75.6** | 69.7 | 53.1 | 61.4 | 62.6 |
| J Decay ↓ | **21.2** | 27.6 | 36.4 | 23.6 | 39.4 |

training sample (first frame in the sequence). From $\mu$, we can get the representation z for the training object, generate the parameters of a segmentation network using the hypernet, then evaluate the segmentation accuracy. We first conduct pre-training on training dataset, acquiring an updated hypernet and new bases. We then test one-shot learning on the DAVIS evaluation set, which contains 20 video sequences. We use the first frame as initialization and evaluate the rest frames. Finally, we compare our results with a set of semi-supervised video object segmentation works on the DAVIS benchmark, using the Jaccard index (IoU) as the evaluation criterion.

Tab. 14 shows the quantitative evaluation on DAVIS. We report Jaccard Mean (average Jaccard score for all test objects), Jaccard Recall (average Jaccard score for test objects whose score is higher than 50.0), and Jaccard Decay (evaluate the Jaccard score decay by time). We compare our framework with video object segmentation methods that use both appearance and temporal information, such as SEA (Avinash Ra-

Table 15: The number of optimizable parameters and average time consumed per object in one-shot learning.

| Measure | ObjP. (Ours) | OSVOS |
|---|---|---|
| # of Parameters | **52** | 5,426,529 |
| Learning Time (s) | **10** | 90 |

makanth & Venkatesh Babu, 2014), HVS (Grundmann et al., 2010) and JMP (Fan et al., 2015). We also compare our work with OSVOS (Caelles et al., 2017), a recent work of one-shot learning on video without temporal information. To make a fair comparison, we implement OSVOS using the same structure as the primary network in our framework, and remove the post-processing. As shown in Tab. 14, our method outperforms these four methods on both Jaccard mean and recall. Specific experiment settings are described in A.9. Generally speaking, our method is comparable to the state-

**one-shot training data**

**tesing data**

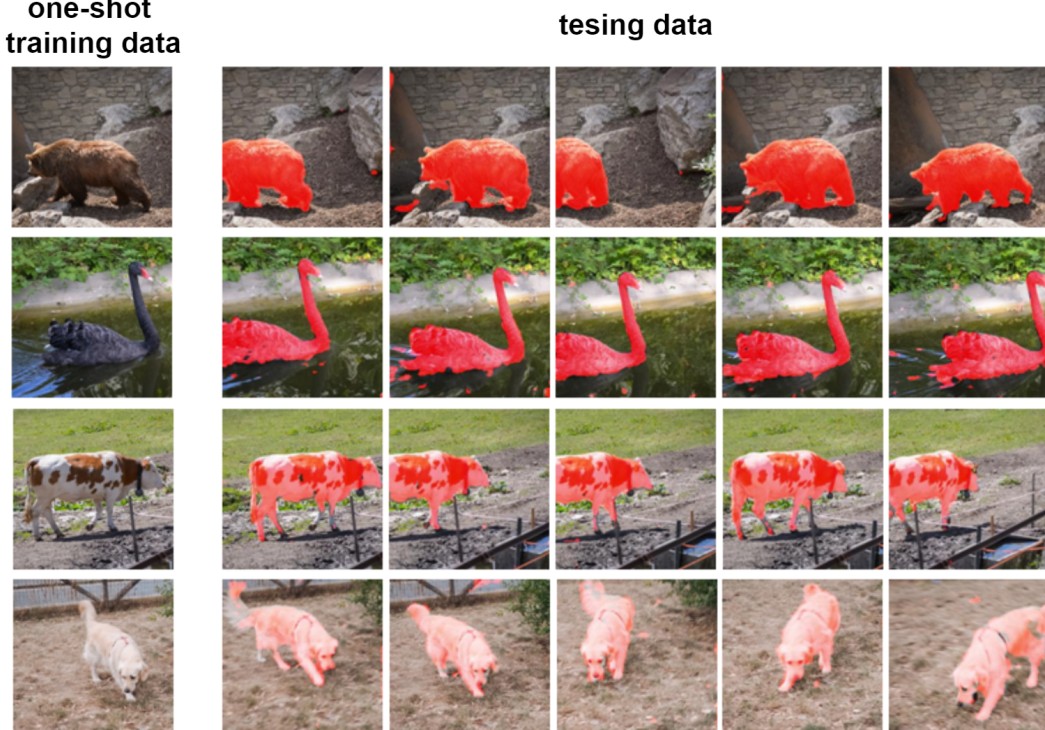

Figure 7: Visualization results of one-shot learning on DAVIS dataset. 1st column: training data that contains only one data sample. 2nd to 6th column: results on testing set (the rest frames in the video clip).

of-the-art on one-shot object segmentation learning, though there is enough space for us to push the performance.

Furthermore, in terms of learning efficiency, our method performs much better, as shown in Tab. 15. Since we only need to optimize the coefficient $\mu$ with the size of $|\mathbf{z}|$, the number of optimizable parameters of our method is much smaller than that of OSVOS. So our framework is more efficient in storage if the trained networks need to be transmitted and used elsewhere. Another advantage of our framework is that the one-shot learning is much faster, which shows the potential of using our framework for real-time applications.

Fig. 7 shows some visualization results of one-shot learning on the DAVIS dataset. Although only one sample is provided during one-shot training, our model can predict the masks on subsequent frames. In some sequences, for example, the 'dog' sequence showed in the 4th row of Fig. 7, the viewing angle and the object's shape vary significantly between frames, making it challenging to predict subsequent frames only based on the first frame without any object prior. It could be inferred from the results that our model contains useful object-centric priors that help segment objects in subsequent frames.

### A.9 DETAILS OF REAL-WORLD ONE-SHOT LEARNING

To achieve better performance on one-shot learning tasks, we explore one-shot learning on real-world datasets. We find several critical factors that would significantly affect the one-shot learning performance from these experiments. Section A.8 shows the final result we acquire on one-shot learning. In this section, we'll show how we reach the final score step by step.

#### A.9.1 PRETRAINING: STREAMED LEARNING V.S. JOINT LEARNING

We propose two ways to learn objects' representation: streamed learning and joint learning. In streamed learning, objects are learned one by one in a certain order, as the algorithm algo.1 shows. To prevent catastrophic forgetting problems in streamed learning, we add memory constraints with a coefficient $\beta$, as Eq. 4 shows. In joint learning, objects are learned together. Specifically, suppose

we have $n$ objects in total, we jointly update the hypernet and $n$ representations ($z$) during training. Each mini-batch of training data contains data of a randomly selected object. Different objects appear alternately, and the representation $z$ corresponds to the current object is updated. Forgetting problems would not occur in joint training.

In this experiment, the hypernet predicts the parameters of the whole segmentation network. We train it on the DAVIS training set, which contains 30 objects. The experiment found that 'joint learning' reaches higher average segmentation accuracy (52.96) than streamed learning (49.72). There're two possible reasons. First, we find that the average validation accuracy of joint training (85.6) is higher than streamed learning (81.2), suggesting that joint learning acquires better base representations that would help represent a novel object. Second, although the memory constraint of streamed learning helps the model remember previous knowledge, it also restricts the searching range, which would cause negative effects on finding representation for a novel object. We'll use joint learning for pre-training in the following experiments to reach better performance.

### A.9.2 NETWORK ARCHITECTURE

As the experiment section 4.1 introduces, we use an encoder-decoder network as the primary network (segmentation network). In the last experiment (A.9.1), the hypernetwork predicts the parameters of both the encoder and the decoder. However, there's another option: the hypernetwork predicts the parameters of the decoder only. In this setting, the encoder updates with the hypernetwork in pre-training, and is fixed in one-shot learning.

In this experiment, we find that such architecture differences would significantly affect the one-shot performance. For pre-training, we use 'joint learning', as we mention in A.9.1. Other settings are the same as A.9.1. When the hypernetwork predicts the decoder only, we find that the test score (58.95) is much higher than predicting the whole network (52.96). In our implementation, the encoder has more network parameters than the corresponding hypernet that generates its parameters. The hypernet could only generate a subset of all possible parameters of the encoder. The limitation of the searching range in parameter space may cause the output result to fall into local extrema. Since the encoders under these two settings are both fixed during one-shot learning and share the same structure, we assume that an independently trained encoder that is not predicted by the hypernet can better extract useful high-level features for the downstream processing. We'll use this setting in the following experiments.

### A.9.3 PRETRAIN DATA

In our previous experiments, we train the hypernet and object representations on the DAVIS training set, which contains 30 objects only. In this section, we expand our training dataset with Youtube VOS (Xu et al., 2018). In total, we train our network with 694 real-world objects; each object contains a 60-90 frames video sequence. Compared to the previous test score (58.95), training on our extended dataset reaches a much higher score (64.45). When our model is trained on more objects, it learns more prior knowledge about objects, thus performing better on the one-shot learning task. Training with more objects would help our model better discriminate the shared object properties that could be learned by the hypernet and the independent properties that should be saved in the object's representation $z$.

We also conduct experiments to evaluate the effect of the bases. In previous experiments, we represent novel objects over base object representations; in this part, we also perform learning of the object representations over the entire representation space. Tab.16 shows that

Table 16: One-shot learning accuracy and training efficiency. One-shot training is performed by searching the optimal representation either on the manifold spanned by the base objects, or over the entire representation space.

|  | base representation | full representation space |
| --- | --- | --- |
| J Mean ↑ | 64.45 | 48.48 |
| Learning Time (s) ↓ | 12 | 103 |

searching over base objects representation space performs much better than searching over full representation space on both testing accuracy and training efficiency. We show a similar pattern in 4.2.3. In this experiment, we construct the base representation space through joint training on 694 object data. In the 'full representation space' setting, we use the same hypernet as the 'over base

representation' setting. The result probably suggests that the base representation space, as a sub-space of the full representation space, contains objects' prior knowledge and shares some common information about objects. When a novel object comes, it is more likely to accurately represent it in this sub-space than in the full representation space. Searching over the full representation space would make the searching process slower and possibly fall into local maxima.

### A.9.4 FINAL RESULT

The final result is shown in Tab.17. The best score is acquired when we pre-train our model on 694 objects (YoutubeVOS + DAVIS) using joint training, and the hypernet predicts the parameters of the decoder only. The Jaccard score increases from 49.7 to 64.5 by modifying the critical factors.

Table 17: Jaccard index on DAVIS evaluation set.

| Measure | ObjP.(Origin) | ObjP.(Now) |
|---|---|---|
| J Mean ↑ | 49.7 | **64.5** |
| J Recall ↑ | 62.3 | **75.6** |
| J Decay ↓ | 29.4 | **21.2** |

