# OpenReview forum: "Object Pursuit: Building a Space of Objects via Discriminative Weight Generation"
_ICLR.cc/2022/Conference — ICLR 2022 Poster_

### Official Review · Reviewer_jB2Q · 2021-10-20

**Correctness:** 3
**Technical Novelty And Significance:** 3
**Empirical Novelty And Significance:** 2
**Recommendation:** 6
**Confidence:** 2

**Main Review:**

Strengths:

1. The idea of using the network weights to represent object-centric representations seems new and interesting.

2. In the experiment section, the authors present an extensive study to understand the presented methods.

Weaknesses:

1. The method requires a semantic segmentation mask as input even during testing. This largely limits the application scope as annotating mask even for an image is quite difficult.

2. All experiments in the manuscript are conducted with synthetic data, which is not that convincing. In the supplementary material, the authors have presented some real-world examples. However, it still lacks qualitative results.

**Summary Of The Paper:**

This paper presents a new framework to learn object-centric representations. The model is composed of a segmentation network and a hypernetwork. The hypernetwork takes the latent representation of a certain object as input and predicts the weights for the segmentation network. The latent representation and the hypernetwork are jointly optimized to maximize the discrimination power of the segmentation network. The framework also introduces a sparse representation mechanism so that each object can be represented in some base object representations.

**Summary Of The Review:**

Overall, I think the basic idea of leveraging network weights for object-centric representation is interesting. However, I am quite doubtful about its practical value because of the need of a mask during testing and the lack of real-world experiments.

---

> ### Author Response · Authors · 2021-11-23
> **clarifications and real-world results**
>
> Thanks very much for your time and valuable suggestions. We are glad to see your appreciation of the novelty and the extensiveness of our experiments. Next, we address the questions raised in the reviews.
>
> **Q1.** The method requires a semantic segmentation mask as input even during testing. This largely limits the application scope as annotating masks even for an image is quite difficult.
>
> **A1:** We think this may be a misunderstanding. To clarify, we do not use semantic segmentation masks in any case, and the whole pipeline can run without semantic segmentation masks. Specifically, during training, our method needs instance masks of the object being manipulated. Moreover, obtaining the instance masks can be done via motion segmentation, as demonstrated in Section A.5.1 and Fig. 6 in the revised paper. We hope this clarification can help resolve this concern.
>
> **Q2.** All experiments in the manuscript are conducted with synthetic data, which is not that convincing. In the supplementary material, the authors have presented some real-world examples. However, it still lacks qualitative results.
>
> **A2:** Thanks. In section A.5.2, we conduct experiments on real-world datasets, such as Youtube VOS and CO3D. For each dataset, we collect a large number of objects to examine the performance of our framework, specifically 65 instances from Youtube VOS and 285 instances from CO3D. The number of training objects we use from the real world (350) far exceeds the number we collect on synthetic data (60). Moreover, we report quantitative results on the DAVIS datasets to complement the qualitative results, shown in Section A.8, Tab. 14 and 15.
>
> We perform the major experiments on synthetic data due to the lack of a robot to perform the data collection in the real world. Moreover, we like to have a proof of concept before carrying out real-world experiments, which could involve an enormous amount of research funds and effort, which is out of the scope of our current work. However, we are confident that transitioning from the synthetic environment to the real world is highly practical as all the technical components that need to be used in the real world are ready as discussed in A.5.1 and A.5.2.
>
> Please check these experiments on real-world data and let us know if they alleviate your concern. Even though our contribution is the proposed object-centric representation and the (interactive) continuous learning framework, we would like to perform more experiments on real-world data if you see fit.
>
> Thanks for your time again, and we hope that our reply can help resolve your concerns. If this is the case, we sincerely hope that you raise the score accordingly. Please feel free to let us know if you have further questions.

---

> > ### Comment · Reviewer_jB2Q · 2021-11-30
> > **Response to the authors**
> >
> > Thanks for the clarification and the real-world results. They've addressed my concerns, and thus I've raised the score.

---

### Official Review · Reviewer_M8h3 · 2021-11-01

**Correctness:** 2
**Technical Novelty And Significance:** 2
**Empirical Novelty And Significance:** 2
**Recommendation:** 5
**Confidence:** 3

**Main Review:**

- Strength: The authors conduct many ablation studies to analyze the effects of different components of the proposed framework and discuss the role of many factors in-depth, such as influences of hyperparameters and order of training orders.

-- Weakness:

- Model correctness and effectiveness:
(1) The solution to equation (3) seems not to exist theoretically. For positive alpha value (see Training details), since the neural network Phi contains linear transformation operation, the L1 norm for mu can be sufficiently large to increase the value of the objective function.
(2) In my opinion, for positive alpha (see Training details), maximizing the L1 norm of mu would not be a regularizer to encourage sparsity of mu, as mentioned by the authors.
(3) Since one image usually contains multiple objects types or numbers, will it result in misidentification in the object pursuit process? This influence is not discussed in the manuscript.

- Model evaluation:
(1) The model is only quantitatively evaluated on a synthetic dataset with a limited number of objects, lacks quantitative evaluation on a real dataset.
(2) The author argues that the proposed object pursuit framework shows potential for tasks at a low-annotation regime. However, the study lacks comparison with any other models developed for reducing annotation costs. It greatly limits the contribution of the study.
(3) It is evident that the variation of the order of training objects would result in different trained models. Even though the author presents Table 5 to demonstrate this issue, it lacks statistical testing to conclude that the pursuit process is robust to the training object order.

- Writing issue:
(1) For equation 1, the symbols for sample and population are not distinguished.
(2) The meaning of n and m in 3.2 needs to be explained. Do they refer to types of objects and the number of learned objects?
(3) The meaning of tau2 in the A.1 algorithm is not defined and explained in section 3.2 in the manuscript.
(4) Does the alpha in equation (3) and equation (4) refer to the same alpha?
(5) In Section 4.5, (Fig. ??) refers to which figure?
(6) In section 4.3, table3 should be table5?
(7) what does the symbol |*| mean in equation (2)? Does it refer to the number of learned objects? as well as the |*| in equation (3)


- Other issues:
(1) Should the mu be replaced by z in equation (2)? It seems that mu does not exist in the objective term.
(2) For parameters alpha and beta in section 3.2, are they positive or negative numbers? They were set positive in Training details but were claimed to be negative in section 3.2.


**Summary Of The Paper:**

This paper proposes a framework to continuously learn object representations and has analyzed the framework’s feasibility for segmentation task at a low annotation cost on a synthetic dataset. Experiments were conducted in an attempt to demonstrate its effectiveness in representing objects and its potential for label efficient visual learning.

**Summary Of The Review:**

Due to insufficient comparison with other related studies, lack of evaluation on real image datasets, and less rigorous in theory, this research still needs a lot of improvement to meet the accepted standard.

---

> ### Author Response · Authors · 2021-11-23
> **further clarifications and evaluations**
>
> Thanks very much for your time and valuable suggestions. We are glad to see your appreciation of our extensive ablation studies and in-depth discussions of different learning factors. Next, we address the questions raised in the reviews.
>
> **Q1.** Model correctness. 1)The solution to equation (3)... For positive alpha 2) for positive alpha (see Training details), maximizing the L1 norm of mu ...
>
> **A1:** Thanks for pointing this out. Sorry for the confusion caused by omitting the sign in the training details. Alpha and beta should be negative numbers as described in the text below Eq. 4, “The value of the negative scalar coefficients α, β will be detailed in the experiments.” We have corrected the sign in the training details.
>
> **Q2.** Effectiveness: Since one image usually contains multiple objects types or numbers, will it result in misidentification in the object pursuit process?
>
> **A2:** The object pursuit process considers objects that are currently under manipulation (a single object is being manipulated at a time), so the ambiguity is alleviated through interaction, i.e., hand-directed attention can be calculated through calibration of the robotic arms; also motion can be used to obtain the instance masks of the manipulated object. A real-world experiment is shown in Section A.5.1 and also in Fig. 6 of the revised paper.
>
> **Q3.** The model is only quantitatively evaluated on a synthetic dataset with a limited number of objects, lacks quantitative evaluation on a real dataset...comparison with any other models...
>
> **A3:** We conduct further experiments on real-world data, from real-world object pursuit to its one-shot learning application. For real-world object pursuit, we use the YouTube VOS dataset and the CO3D dataset, with more than 300 objects in total. This quantity of objects can simulate a real scene with tons of objects in it. We show the pursuit dynamics and learning efficiency of our framework in Section A.5.2, to validate the potential that our framework can be applied in real scenes. As observed, the conclusions we have obtained from the synthetic data can transfer well to real-world data. For one-shot learning on real-world data, we show quantitative results and compare our work with related works in Section A.8. It shows that our method and the trained model is on par with state-of-the-art one-shot video object segmentation (OSVOS), while our method is much more efficient in terms of storage and one-shot learning time (Section A.8, Tab. 15), which demonstrates the potential of our method to reduce annotations in practice. Again, our main contribution is the proposed object-centric representation and the object pursuit framework to learn objects in a continuous way with reidentification and forgetting prevention.
>
> Currently, we perform the major experiments on synthetic data due to the lack of a robot to perform the data collection in the real world. Moreover, we like to have a proof of concept before carrying out real-world experiments, which could involve an enormous amount of research funds and effort, which is out of the scope of our current work. However, we are confident that transitioning from the synthetic environment to the real world is highly practical as all the technical components that need to be used in the real world are ready.
>
>
> **Q4.** It is evident that the variation of the order of training objects would result in different trained models. Even though the author presents Table 5 to demonstrate this issue, it lacks statistical testing to conclude that...
>
> **A4:** This might be a misunderstanding due to a missing referenced table.
> We make a change in Section 4.3 marked in red. Could you please check it and see if this resolves this question?
> In a nutshell, Tab. 5 shows statistical results of pursuit dynamics under random order of training objects. The small std-dev in Tab. 5 shows that the order of training objects will not have a significant impact on pursuit properties, such as the number of bases, the length of the object list, and the average segmentation accuracy. It illustrates that the pursuit process is robust to the object order.
>
> **Q5.** Writing issues and clarifications.
>
> **A5:** We made corrections and clarifications to the issues mentioned above. In particular, we **1)** add subscripts to distinguish symbols in Eq. 1; ** 2)** clarify the meaning of m and n; **3)** clarify that \tau1 and \tau2 in the algorithm is the same; **4)** \alpha in Eq. 3 and 4 is different as stated in the training details; **5)** add missing reference numbers in Section 4.5 and 4.3; **6)** clarify the symbol || before Eq. 2; **7)** explicitly write \mu in Eq. 2; **8)** add a minus sign to \alpha and \beta in training details.
>
> Thanks for your time again, and we hope that our reply can help resolve your concerns. If this is the case, we sincerely hope you can raise the score accordingly. Please also feel free to let us know if you have further questions.

---

> > ### Comment · Reviewer_M8h3 · 2021-12-04
> > **Response to authors**
> >
> > Thanks very much for the clarification. I have read the authors’ response and other reviewers' comments carefully.  Even after rebuttal, the results on real world data are still very limited. I would like to keep my rating.

---

### Official Review · Reviewer_LiL2 · 2021-11-01

**Correctness:** 3
**Technical Novelty And Significance:** 3
**Empirical Novelty And Significance:** 3
**Recommendation:** 6
**Confidence:** 3

**Main Review:**

In this paper, the author proposes a novel framework named object pursuit which can continuously learn object-centric representations with streaming training data.

Generally speaking, the paper is well formulated and well written. This paper has a good formulation. Also, the proposed framework is well illustrated and easy to follow.

However, there are some small issues that the author should address:

1. Some grammar and writing issues: the author should improve their writing. For example, Fig ?? in Sec 4.5 and some other grammar mistakes.

2. Somehow the generated dataset is small. Can the author provide some statistics about the efficiency, i.e., running time/FPS of the proposed framework? I'm a little bit worried about applying it to a large-scale (real-world) task.

**Summary Of The Paper:**

In this paper, the author proposes a novel framework named object pursuit which can continuously learn object-centric representations with streaming training data. The author collected data from interactions and the proposed framework can imporve the label efficiency in downstream tasks.

**Summary Of The Review:**

Generally speaking, the paper is well formulated and well written. This paper has a good formulation. Also, the proposed framework is well illustrated and easy to follow. Some small issues remain but they do not affect the overall quality of the paper.

---

> ### Author Response · Authors · 2021-11-23
> **additional details and experiments**
>
> Thanks very much for your time and valuable suggestions. We are glad to see your appreciation of the novelty and comprehensiveness of the proposed framework. Next, we address the questions raised in the reviews.
>
> **Q1.** grammar and writing issues... For example, Fig ?? in Sec 4.5 and some other grammar mistakes.
>
> **A1:** Thanks for pointing them out. We have made corrections accordingly in the revised version. Please let us know if there are other edits you see fit.
>
>
> **Q2.** Somehow the generated dataset is small. Can the author provide some statistics about the efficiency, i.e., running time/FPS of the proposed framework? I'm a little bit worried about applying it to a large-scale (real-world) task.
>
> **A2:** Thanks for asking for the run-time and statistics. We should have this in the paper (now we have included them in the revised paper). For the synthetic data we collected on iThor, the object sequences are from 67 objects (an extra set of 52 objects for pretraining). The run-time of our framework on synthetic data depends on the threshold $\tau$. Generally, a smaller $\tau$ leads to a shorter running time since objects are easier to be expressed by bases or recognized as seen objects, which will save the time of learning base object representations. In our experiments, $\tau=0.5$ could finish running 67 objects in about 10 hours, while $\tau=0.8$ needs 1.5 days. We also conduct real-world experiments on youtubeVOS and CO3D datasets. Both have a running time of about 60 objects per day under $\tau=0.7$. We collect 65 objects from youtubeVOS and 285 objects from CO3D for the real-world experiments, simulating the large-scale tasks in the real scene. Please also check Section A.5.2 for more details.
>
> We perform the major experiments on synthetic data due to the lack of a robot to perform the data collection in the real world. Moreover, we like to have a proof of concept before carrying out real-world experiments, which could involve an enormous amount of research funds and effort, which is out of the scope of our current work. However, we are confident that transitioning from the synthetic environment to the real world is highly practical as all the technical components that need to be used in the real world are ready, as discussed in A.5.1 and A.5.2.
>
>
> Thanks for your time again, and we hope that our reply can help resolve your concerns. Please feel free to let us know if you see things that we should incorporate or edit.

---

> > ### Comment · Reviewer_LiL2 · 2021-12-01
> > **Response to the authors**
> >
> > Thanks for the clarification, real-world results, and explanations. It partially solves my concerns, but I'm still worried about real-world and large-scale datasets, thus I will keep my current score.

---

### Official Review · Reviewer_SKNa · 2021-11-02

**Correctness:** 3
**Technical Novelty And Significance:** 2
**Empirical Novelty And Significance:** 3
**Recommendation:** 5
**Confidence:** 3

**Main Review:**

This paper is well-organized and tackles the problems that can happen for continuous learning of object representation. They consider how to identify whether an object is seen or unseen, and progressively enrich the base concepts, and introduce redundancy removal, and forgetting prevention.

It is reasonable for setting a \tau in the formulation to identify the unseen objects.  Comprehensive studies and explanations over  \tau are included in the experiment parts. To further elaborate on the impact of \tau, would it be good to show the re-identification recall and precision of the unseen objects with different \tau?

As stated in Table 1, "if an object is learned and added to the object list, it will be claimed as seen by Eq. 2, and the re-identification
accuracy is always one." However, if the unseen objects are added, the unseen objects might be misclassified to some seen objects, and therefore, the precision can be lower than one.

A separate analysis on the objects added and not added to the base class, and their respective segmentation and re-identification performance could be useful to understand how the system works in actual test cases.

The idea of representing new concepts with a set of known concepts and incrementally adding new concepts is not new as it has been used in memory mechanisms and other few-shot learning works.

It is a bit unclear how this can be extended to real-world data where a robot agent needs to obtain the instance segmentation mask through interaction for a large diversity of objects.

**Summary Of The Paper:**

This paper proposed a framework to continuously learn object-centric representations and formulate the problem by projecting the object representation space to the hypernetwork parameters for the segmentation task. The data are sampled from marginals where only one instance mask is collected in each scene. The representation learning incorporates the base representation learning, redundancy removal, and forgetting prevention.  The experiments study the impact of \tau under different metrics.

**Summary Of The Review:**

This paper is a comprehensive pipeline for continuous object learning and presents some valid designs for this specific problem. But it is a bit unclear how this can be extended to real-world data where a robot agent needs to obtain the instance segmentation mask through interaction for a large diversity of objects. Some of the evaluation metrics are not very clear to illustrate the model performance.

---

> ### Author Response · Authors · 2021-11-23
> **additional results and experiments on real-world data**
>
> Thanks a lot for your time and valuable comments and suggestions. We address the questions and requested experiments in the following.
>
> **Q1.** To further elaborate on the impact of \tau...
>
> **A1:** We added the requested table in the appendix (Section A.6, Tab. 11).
>
> | \tau | 0.5 | 0.6 | 0.7 | 0.8 |
> | :--: | :--: | :--: | :--: | :--: |
> | recall  | 0.28 | 0.40 | 0.56 | 0.72 |
> | precision | 1.0 | 1.0 | 1.0 | 1.0 |
>
> As observed, the re-id precision on unseen objects is 1.0 across all \tau’s, indicating that none of the seen objects are identified as unseen, which signals that our forgetting prevention is effective.
>
> On the other hand, the recall is less than 1.0 and decreases as \tau decreases. We have provided an analysis in Section 4.2.1. The reason is that an unseen object will be claimed as seen if a similar object is learned (seen). For example, two very similar apples (also see Fig. 3 in the main paper). This is a desired characteristic since all apples in the box are theoretically different; however, an intelligent agent would not spend time learning every apple. Moreover, if the task is to differentiate each apple from the other, one can increase \tau, so two apples will be harder to claim as similar (Eq. (2)) (also evidenced by the increasing recall as \tau increases).
>
>
> **Q2.** As stated in Table 1, "if an object is learned..."
>
> **A2:** This is true, as observed from Tab. 2. In the following table (Section A.6, Tab. 12), we show the requested results. Now, the recall is still 1.0 across different \tau’s, but the precision decreases when \tau decreases, aligning with our observation in Tab. 2, since unseen objects can be identified as seen if the network has learned similar objects (also discussed in Q1).
>
> | \tau | 0.5 | 0.6 | 0.7 | 0.8 |
> | :--: | :--: | :--: | :--: | :--: |
> | recall  | 1.0 | 1.0 | 1.0 | 1.0 |
> | precision | 0.60 | 0.64 | 0.71 | 0.80 |
>
> In Tab. 1, we want to know if all seen objects can be identified as seen (recall) so that the network does not need to learn the same object repeatedly. At the same time, we like to know if the network can correctly predict the identity of the seen object (re-identification accuracy), which is slightly different from how accurate the seen prediction is.
>
> **Q3.** A separate analysis on the objects added and not added...
>
> **A3:** Thanks for pointing this out. Section A.7 shows the requested metrics.
>
> We observe that both base and non-base objects can be re-identified correctly across different \tau’s. The segmentation accuracy of base and non-base objects increases with \tau. The re-id performance of our framework is stable and accurate on both base and non-base objects.
>
> **Q4.** memory mechanisms and other few-shot learning...
>
> **A4:** We agree that some high-level ideas may sound familiar to works from memory mechanisms and few-shot learning. However, we like to point out that: 1) our representation of objects is fundamentally different. The network weights are the representations instead of feature vectors, making continuous learning of individual objects without forgetting much more efficient. 2) The whole framework is focused on life-long object-centric representation learning from interactions. We have developed a comprehensive pipeline and performed extensive studies as acknowledged. To the best of our knowledge, we have not seen a work similar to ours regarding how object-centric representation is defined and how they are learned continuously. As also acknowledged by Reviewer LiL2: "proposes a novel framework..." and Reviewer jB2Q: "presents a new framework..."
>
> **Q5.** It is a bit unclear.. real-world data...
>
> **A5:** In order to evaluate how practical it is for a robot to perform the pursuit process in the real world, we need to look into two aspects. First, how possible it is to obtain instance masks of the manipulated object. Second, how efficient the learning can be given the objects that need to be learned. We examine these two aspects in Section A.5 in the revised paper and show that they are practically feasible and efficient in the real world. Please refer to the details there.
>
> In a nutshell, we have confirmed that obtaining instance masks of real-world objects via interaction is practical (Section A.5.1). The instance masks can be obtained by a motion segmentation network (well studied) without any human supervision. In practice, the robotic arm can be self-labeled and learned as an object to further improve the instance mask. In Section A.5.2, we experiment on Youtube-VOS and CO3D, consisting of real-world data. With a single GPU, the pursuit can finish 40 objects a day, and this should be more practical than doing manual annotation for all the objects.
>
> Thanks again for your time. We hope our discussions can resolve your concerns, and if that is the case, we sincerely hope that you can raise the score accordingly. In case you have further questions, please also feel free to let us know.

---

### Author Response · Authors · 2021-11-23
**thanks to all reviewers**

We want to thank all reviewers for the constructive comments, suggestions, and acknowledgments on the novelty of the proposed framework, that the paper is well formulated and written, and comprehensive studies and in-depth discussions. We address each of the questions and concerns in the following.

---

### Decision · Program_Chairs · 2022-01-20

**Decision:**

Accept (Poster)

**Comment:**

This paper received 4 quality reviews. The reviewers like the problem formulation and various ideas presented to enabling a working pipeline. However, almost all experiments are conducted on synthetic data. Concerns are also raised regarding its usage and application on real-world data. In the updated version, the authors add some results on real-world data, yet without quantitative evaluation. After the rebuttal and discussion, the final rating is 6 from 3 reviewers, and 5 from 1 reviewer (note that reviewer SKNa stated that the rating will be increased to 6 but did not end up changing it in the "recommendation" entry). The AC sees both pros and cons of this work. Given that a conference paper does not have to be comprehensive in all frontiers and the novel idea presented in this paper, the AC is leaning toward in favoring of this work and thus recommends acceptance.